# Cryogenic Comminution of Subsea Cables and Flowlines: A Pathway for Circular Recycling of End-of-Life Offshore Infrastructure

Ibukun Oluwoye * and Arun Mathew

Curtin Corrosion Centre, Western Australian School of Mines: Minerals, Energy and Chemical Engineering, Curtin University, Perth, WA 6102, Australia
* Correspondence: ibukun.oluwoye@curtin.edu.au

**Abstract:** Hundreds of thousands of kilometers of communication and power (umbilical) cables and flowlines lie undersea worldwide. Most of these offshore cables and flowlines have reached or will soon be nearing the end of their service life, prompting the need for a viable recycling approach to recover some valuable material, e.g., copper. However, separation into constituent materials has proven very challenging due to the highly robust design of the composite cables (and flowlines) to withstand service conditions and the tough external plastic sheaths that protect against seawater corrosion. This study aims at promoting sustainable practices in the offshore energy sector. Here, we summarize the findings of the cryogenic comminution of subsea cables and flowlines for an effective separation and recovery of component materials. Heat transfer analyses of complex multilayer flowlines and umbilicals were conducted to evaluate the time required for these structures to reach their respective critical brittle-transition temperatures. Subsequently, the time was used as a guide to crush the flowline and umbilical cables under cryogenic conditions. The results show that the flowlines and umbilical cables will reach the brittle-transition temperature after approximately 1000s (i.e., 17 min) of submergence in liquid nitrogen (LN). Comminution of the materials at temperatures near the brittle-transition temperature was proven relatively efficient compared to room-temperature processing. The present evaluation of heat transfer and lab-scale crushing will afford accurate process modelling and design of a pilot cryogenic comminution of decommissioned subsea cables and flowlines, enabling the sustainable recovery of valuable materials that can provide a new stream of waste-to-wealth economy.

**Keywords:** material recycling; subsea cables; offshore infrastructures; flowlines; decommissioning; end-of-life



## 1. Introduction

Designing an appropriate waste management option that can accommodate end-of-life subsea communication and power (umbilical) cables and flowlines (in addition to Supplementary Parts) remains one of the prime foci of offshore energy sectors [1–5]. This initiative is vital because an adequate recycling opportunity will provide environmental and cost benefits to offset the estimated financial liability of removing various forms of production infrastructure from depleted offshore oil and gas fields [3,6].

There are over 12,000 offshore installations globally [7] across the continental shelves of about 53 countries [5], primarily being managed by the offshore renewable energy, oil and gas, and maritime sectors. In addition, there are currently about 380 underwater cables worldwide, spanning a length of over 1.2 million km [8]. To put it into perspective, the length of these submarine cables is enough to circle the earth more than 25 times. Moreover, the total number of these cables changes constantly due to new offshore developments and the decommissioning of end-of-life offshore infrastructures. While these offshore infrastructures produce different environmental impacts [9,10], proper management of their

decommissioned wastes remains crucial for the circular economy through the recovery of valuable metals and non-metals. From an environmental viewpoint, the synthetic polymer and plastic components should be properly managed to avoid persistent contamination of the environment in the form of micro- and nano-plastics. This is so because many offshore infrastructures are ladened with plastics for (i) protection against seawater corrosion, (ii) thermal insulation, (iii) electrical insulation, and (iv) inner protection and paddings [11].

Separating the cables and flowlines into constituent materials has proven challenging due to the highly robust armored design (Figure 1) of the composite cables (and flowlines) to withstand service conditions, and the tough external plastic sheaths that protect against seawater corrosion.

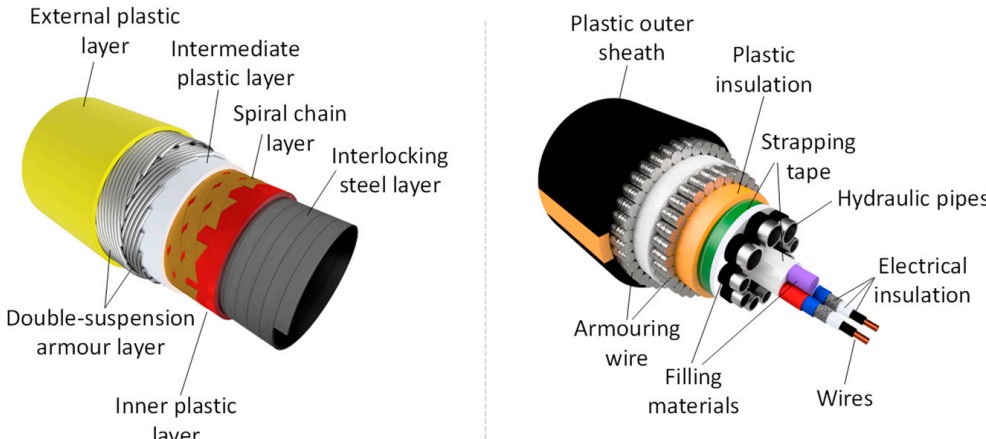

**Figure 1.** Structural examples of subsea flexible flowline (**left**) and umbilical cable (**right**).

Therefore, we envisage development capabilities to tackle this problem and transform decommissioning waste management worldwide. We proposed breaking down the decommissioned end-of-life subsea flowlines and umbilical cables by cryogenic comminution. In addition to cost-effectiveness, the benefits of such a cryogenic comminution process include (i) relatively good comminution efficiency as the materials will exhibit brittle state properties and (ii) capacity to achieve small size distribution (hence, effective separation) of multicomponent materials. This study evaluated the feasibility of cryogenic cooling of subsea flowlines and umbilicals before crushing. Therefore, the objectives include:

(i) a multiphysical modelling of heat transfer and structural transformation of flexible flowlines and umbilical cables under cryogenic environments of liquid nitrogen to determine the critical process parameters such as the transient ductile-to-brittle transition time and glass transition time at a constant cryogenic temperature.

(ii) a two-stage cryogenic crushing test, enabling us to demonstrate the crushing and separation efficiencies, as well as the cost benefits of the process.

This work focuses on heat transfer analyses and lab-scale comminution testing in conjunction with an economic analysis of the cryogenic process. An effective comminution of these subsea cables and flowlines will enable the sustainable recovery of valuable metals and non-metals embedded in the subsea infrastructures and promote circular economy rather than landfill (and other environmentally polluting) disposal options.

## 2. Materials and Methods

### 2.1. Sample Details

The flexible flowlines were provided by oil and gas operators in Australia. The material of a flexible flowline is generally made up of two categories: steel materials that provide mechanical strength and plastic materials that protect the steel and prevent leakages [12]. Details of the flowline design used in this study are shown below. The external plastic layer protects the metal layer from external corrosion and abrasion and bonds the tensile

strip of the base. This layer is often followed by a double-suspension armour layer made of flat steel to prevent tension and resist axial force and internal pressure. Subsequently, the intermediate and internal plastic layers prevent the collapse of the internal sheath (e.g., when the outer sheath is damaged) and seal the pipe to resist wear and internal corrosion. The innermost interlocking steel tire transports the hydrocarbon fluid.

The umbilical cables, also provided by oil and gas operators in Australia, are a composite structure of tube bundles made of various materials [12]. The dimensions of each tube are illustrated below. In general, the design consists of an outer (plastic) protection layer, armoured wires, an internal protective layer, hydraulic pipes, polymer (high hardness) fillings, and electric cables.

### 2.2. Theories

A material may undergo ductile or brittle fracture modes (i.e., the two significant types of fracture) depending on its mechanical properties, type of loading (static, cyclic, strain rate), presence of pre-existing cracks or defects, and environment and temperature. Ductile fracture involves plastic deformation of the material at the crack tip. This often results in a stable and predictable mode of fracture in which crack growth can only occur under an increasing applied load. On the other hand, brittle fracture is the sudden and rapid failure of a material that shows little or no plastic strain at the failure. This is characterized by quick failure without any warning. The generated cracks propagate rapidly, and the material collapses suddenly. Accordingly, brittle materials have relatively lower toughness (i.e., absorb less energy to fracture) due to the relatively smaller area under the stress–strain curve, compared to ductile materials [13]. Therefore, cryogenic processing (via ductile-to-brittle transition) provides the means of lowering the comminution energy of ductile materials.

The ductile-to-brittle transition temperature, often referred to as DBTT, is the temperature at which there is a pronounced decrease in a material's ability to absorb the force without fracturing [13,14]. At this temperature, the material fracture mode transitions from ductile to brittle, requiring a relatively low impact energy. A ductile fracture mostly consumes a high amount of energy during the progress of fracture. Since the toughness of the ductile materials is higher than the brittle ones, the required energy for the failure of ductile materials is usually more than the necessary energy for the brittle materials.

The analogous temperature in polymer materials is referred to as the glass transition temperature ($T_g$), characterizing a considerable decrease in fracture toughness [15,16]. Table 1 lists the DBTT and $T_g$ of the primary components of flexible flowline and umbilical cables; the objective of the cryogenic processing is to initiate such transitions to enable brittle fracturing of the materials.

**Table 1.** The ductile-to-brittle (or glass transition) temperature of some materials in subsea cables and flowlines.

| Material | Ductile-to-Brittle (or Glass) Transition Temperature | Ref. |
|---|---|---|
| Stainless steel | −30 °C to 40 °C | [17–19] |
| Carbon steel [1] | −75 °C to 10 °C | [20,21] |
| Copper | N/A | [21] |
| Polyamide | 35 °C to 50 °C | [22] |
| Polyethylene | −120 °C | [22] |
| Polypropylene | −10 °C | [22] |

[1] Low-carbon and mild steel.

### 2.3. Numerical Simulation and Laboratory Testing Procedures

The convective heat transfer is governed by Newton's law of cooling and was facilitated by liquid nitrogen (LN). The numerical analyses were carried out using a commercial finite volume Multiphysics modelling software package, ANSYS Version 2020 R1, and crushing was performed as described below. The thermophysical properties and polyno-

mial functions to describe the thermal properties of the relevant materials were sourced from the ANSYS material database [23] and the literature [24].

## 3. Results and Discussion

### 3.1. Thermal Analysis of Flexible Flowlines in Cryogenic Environments

The flexible flowlines model created for the thermo-structural analysis is given in Figure 2. The diameter of each component is also shown. The length of the flowlines is considered 100 mm. The computational domain was generated using commercial software Design Modeler 2020 R1, while the mesh was generated using Ansys Meshing software Version 2020 R1. The sweep meshing strategy with regular hexagonal elements was used to generate the mesh. The number of elements created is 24,192.

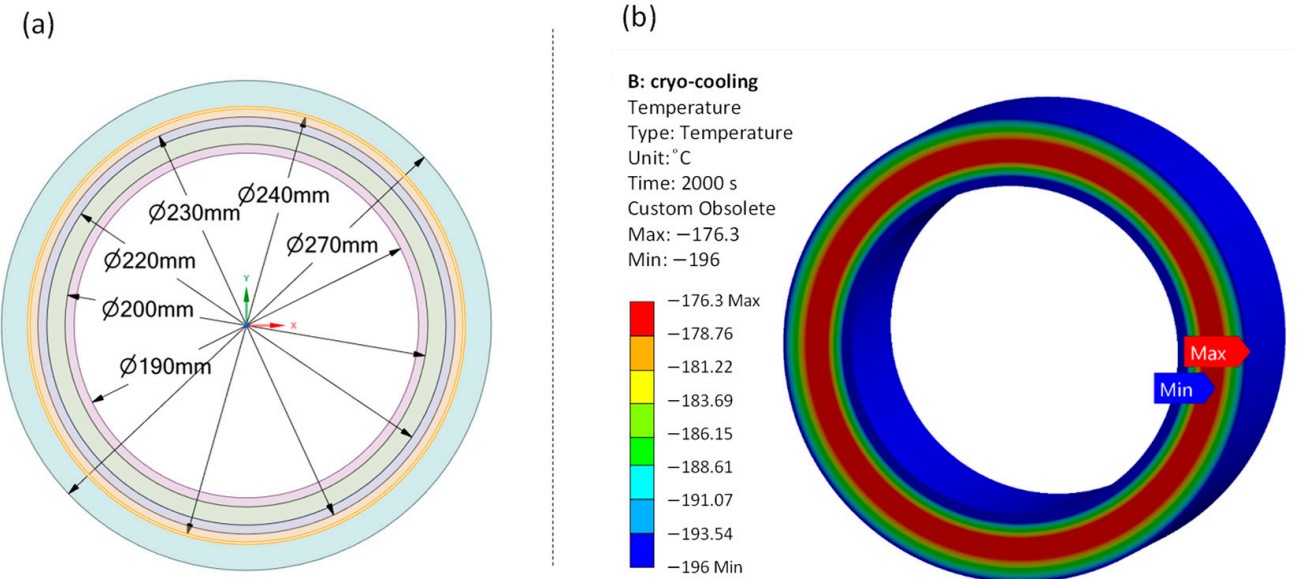

**Figure 2.** (**a**) Schematic of the flexible flowlines cross-section. The flowline comprises five layers: the inner layer (layer 1) is constructed of stainless steel (SS 304). The layer next to the inner layer (layer 2) is made of polyamide, and it is followed by two layers of carbon steel layers (layers 3 and 4), and the outermost layer (layer 5) is made of high-density polyethylene (or polypropylene). The length of the flowline considered in the modelling is 100 mm. (**b**) Temperature contour of flexible flowlines at 2000 s.

The flexible flowline is assumed to be exposed to liquid nitrogen in a relatively large (stationary) reservoir. A boiling heat transfer can be envisaged at the outer solid–liquid interface in this condition. During the boiling heat transfer process, the convective heat transfer coefficient is typically in the range of 10,000–50,000 W/m$^2$·°C. However, because the temperature gradient in this situation is considerably larger than the nucleate boiling zone, a vapour region will form at the solid–liquid interface, which hinders a faster rate of heat transmission to some amount. Previous research found that the heat transfer coefficient during liquid nitrogen chilling ranged from 300 to 800 W/m$^2$·°C [25,26]. The boiling heat transfer coefficient values are also affected by factors such as surface roughness, the presence of contaminants, and the direction of the flowlines in the liquid nitrogen bath [27–30]. As a result, convective heat transfer coefficients of 300, 500, and 800 W/m$^2$·°C are selected for thermal analysis in this work. The flowline's initial temperature is set to 22 °C (i.e., room temperature). Convective heat transfer is assumed at the inner and outer surfaces of the pipe. The surrounding temperature of the flowline is considered to be −196 °C, which is the temperature of LN.

The temperature drop in the flexible flowlines during cryogenic cooling is also depicted in Figure S1 at different heat transfer coefficients. The graph shows that the variable heat transfer coefficient has little effect on the average temperature of the structure under cryogenic conditions. Consequently, in the following section of the simulation, the convective heat transfer value of 500 W/m$^2$·°C is employed. Furthermore, the graph shows that the temperature of various flowline sections drops quickly during the early cooling phase and reaches liquid nitrogen levels in the 2000 s. This ensures that each material in the flowlines reaches a temperature below its ductile-to-brittle (or glass) transition temperature (see Table 1). Furthermore, Figure 2b depicts the temperature contour of the flexible flowline at 2000 s. The minimum temperature is represented at the solid–liquid boundary, while the highest temperature (ca. −176 °C) is shown in the flowline's layers 3 and 4.

### 3.2. Thermal Analysis of Subsea Cables in Cryogenic Environments

Figure 3 shows the umbilical cable model that was used in the thermal simulation. It should be noted that the geometry of umbilical cables varies significantly depending on their types. Therefore, we selected a type randomly for the thermal analysis. The length of the umbilical models is 100 mm. The computational domain was also generated using commercial software Design Modeler 2020 R1, with ANSYS Sweep meshing of hexagonal elements. A total of 135,816 elements are generated after meshing. The initial temperature of the umbilical is considered at 22 °C. A convective heat transfer coefficient of 500 W/m·°C is employed on the outer and inner surfaces of the holes.

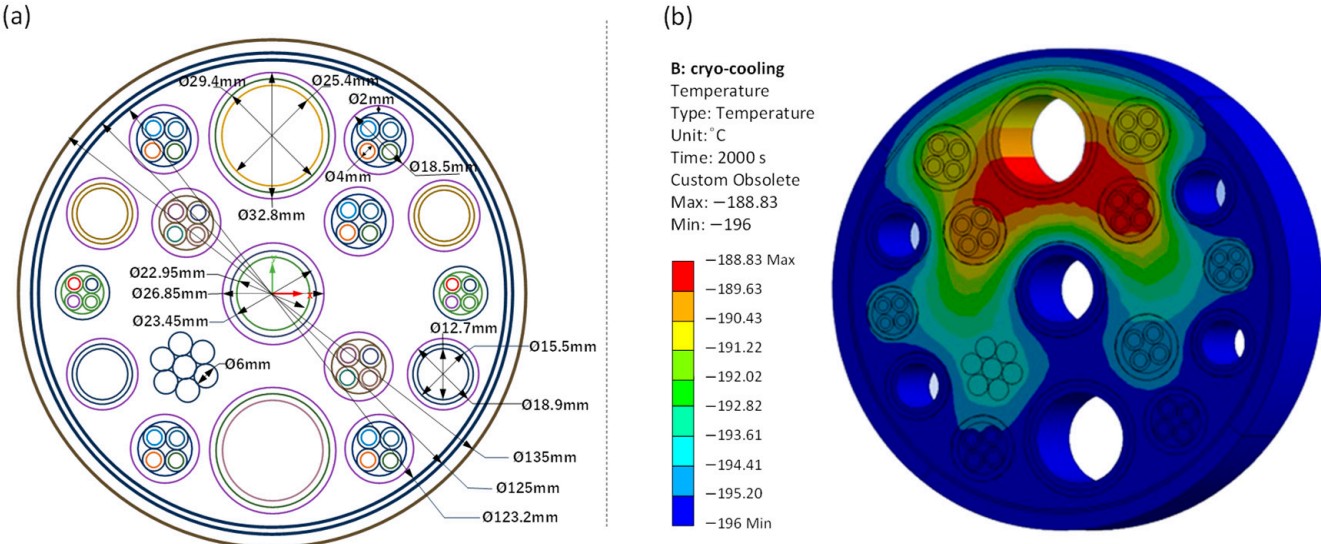

**Figure 3.** (**a**) Schematic of umbilical cable cross-section with an outer diameter of 135 mm. The enlarged version is available as Figure S2. Outer sheath: polyethylene; internal plastics: polyamide; filling material: epoxy thermoset; electrical insulation: PVC, etc. (**b**) Temperature contours at 2000 s internal temperature.

Figure 3b shows the temperature contour of the umbilical cables after 2000 s in the cryogenic cooling process. As expected, the temperature of the umbilical drops rapidly in the initial phase of cooling, and the rate of temperature gradually reduces as the cooling time progresses. After 2000 s, the temperature of each element in the cable falls below the ductile to brittle (or glass) transition temperature, ensuring the umbilical reaches the brittle state. The highest temperature (ca. −188 °C) is considerably below the DBTT of the materials in the system. This demonstrates that the materials have transitioned into a brittle state after 1000–2000 s of exposure to LN.

### 3.3. Two-Step Cryogenic Crushing Test

Figure 4 describes the two-step cryogenic crushing process. The first step involved lowering the specimens (i.e., flexible flowlines and umbilical cables) into a liquid nitrogen tank for a specific time based on the thermal modelling results. The cryogenically cooled materials, now exhibiting brittle characteristics, were then transferred to the hydraulic press for crushing. The controlled experiment was performed with flexible flowlines without cryogenic cooling.

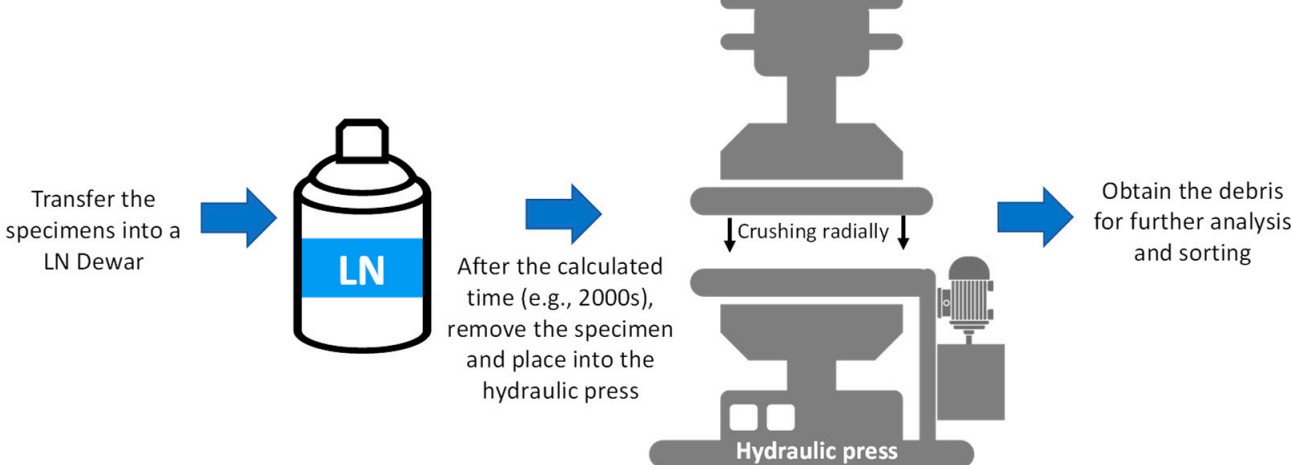

**Figure 4.** Procedure for the two-step cryogenic crushing process. The temperature drops in the material placed in the LN was not monitored experimentally; we relied on the computed time based on ANSYS modelling.

As shown in Figure 5, the cryogenic treatment of the samples resulted in brittle fracturing, and the components of the flexible flowlines and umbilicals were easily liberated from the core steel. Moreover, the control experiment shows that the flexible flowline will not fracture without an initial cryogenic cooling step. We observed the following:

- Waste quality: The quality of the plastic components of the flowline and umbilical cables is still good except for some levels of surface oxidation. The nature and purpose of the various additives should be characterised to find the optimum supply chain for reuse.
- Waste sorting: Plastic wastes are usually sorted through a sequence of sorting steps [31]. Additives can be removed by using gravity in airflow (air classifier), and plastics and metals can be separated by a sink-float density separation system, the magnetic attraction of ferrous metal or by induced magnetic repulsion of nonferrous metals, and a standard IR detector complemented by hyperspectral imaging spectroscopy (HIS).

Our result shows that the component materials can be separated. Moreover, further sorting can be aided by metallurgical techniques such as density separation and flotation techniques.

Furthermore, the cryogenic comminution process can be improved to achieve relatively smaller particle size distribution by introducing one-step techniques. Such a method implies that the LN will be introduced directly into the crushing process to eliminate the heating encountered while transferring the materials into the crusher in a two-step process (see Figure 4).

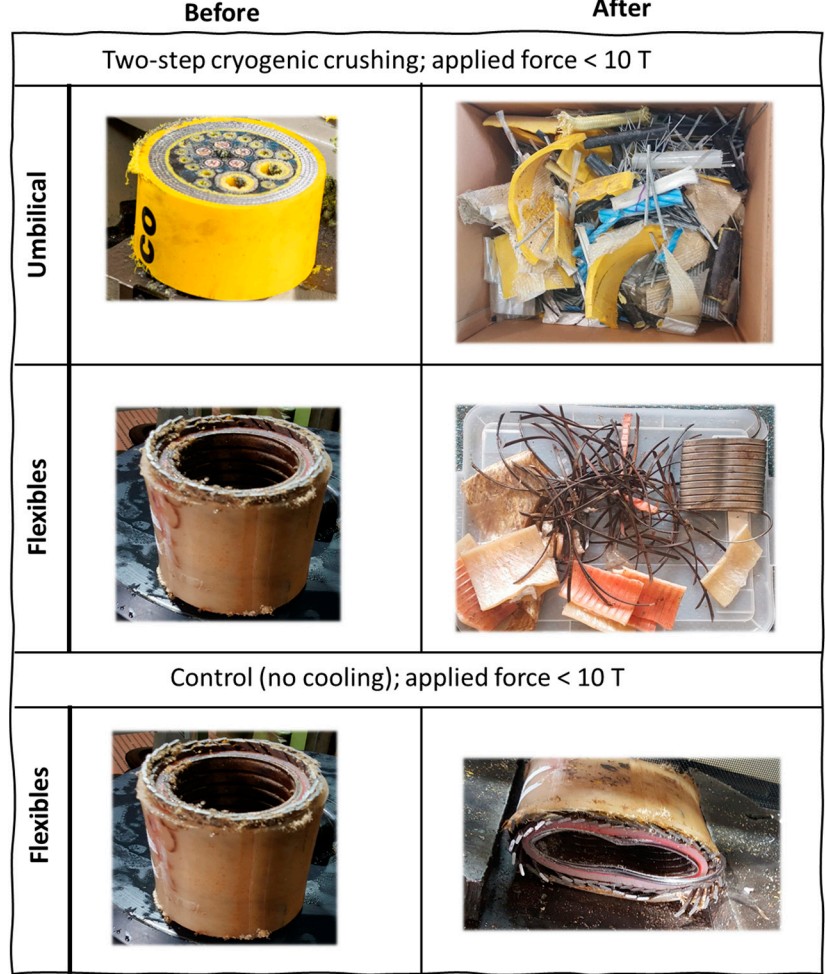

**Figure 5.** Image of the specimen before and after cryogenic crushing. The symbol T represents tonnes. The control experiment was performed without cooling, indicating no visible fracture.

### 3.4. Basic Cost Assessment

We evaluate the cost of employing cryogenic crushing to process flexible flowlines and umbilicals. A proper analysis should be conducted on a pilot scale to demonstrate its financial viability. Therefore, this analysis excludes capital costs (i.e., such as machinery, electricity, technicians, etc.) and focuses mainly on the cost of cryogenic cooling. From a technicality viewpoint, the utilization of LN exhibits application advantages such as:

- Availability: can be easily generated onsite via a cryogenic compressor.
- Handling low technical input: industrial application of LN is readily matured and can be easily controlled.
- Heat transition: conductive and convective mechanisms of LN can be easily managed compared to other cryogens.
- Inert atmosphere: the inertness of LN will preserve the quality of the processed material without any risk of fire. However, this also requires some risk management of the operator's asphyxiation.

On the other hand, economics can be estimated based on the quantity and cost of *LN* required. The theoretical heat needed to cool the materials is the same as the total heat released by the *LN* [32]:

$$Q_{LN} = Q_{feed} + Q_{mill} \qquad (1)$$

The heat introduced by the drive capacity of the mill ($Q_{mill}$) is zero, i.e., the heat from the crusher does not contribute to the cooling in the two-step cryogenic crushing process.

Furthermore, the cooling performance of the liquid nitrogen comprises the condensation and the sensible heat, where $\Delta H_V$ is the enthalpy of evaporation of $LN$ (kJ/kg).

$$Q_{LN} = m_{LN}\left[\Delta H_V + c_{p,LN}\left(T_{f,LN} - T_{in,LN}\right)\right] \tag{2}$$

The theoretical minimum amount (i.e., $m_{LN}$) of $LN$ required for the cryogenic crushing can then be estimated as:

$$\frac{m_{LN}}{m_{feed}} = \frac{Q_{feed}}{\Delta H_V + c_{p,LN}\left(T_{f,LN} - T_{in,LN}\right)} \tag{3}$$

The $Q_{feed}$ cannot be simplified based on the initial temperature of the material $T_{in}$, final temperature $T_f$, mass $m_{feed}$ and heat capacity $c_{p,feed}$; $Q_{feed} = m_{feed} \cdot c_{p,feed}\left(T_{in} - T_f\right)$ because (i) the flexible flowlines are composites of multilayers of different materials, and (ii) in addition to the multiplicity nature, the umbilical cables have complex geometries. Therefore, $Q_{feed}$ is either estimated from the finite element analysis, e.g., by Ansys software Version 2020 R1, or preferably, $m_{LN}$ can be measured experimentally.

We measured the mass of $LN$ after each cooling process to calculate the amount of LN that vaporized during the process. The vaporized content represents the actual $m_{LN}$ which should be re-compressed into $LN$. Accordingly, the cost of cryogenic cooling is equal to the cost of re-compressing the vaporized $m_{LN}$ back into liquid form. Table 2 summarizes the cost comparison. By employing some key technical and cost indicators for the techno-economical assessment, we determined the functional cost of the product materials as compared to the current market prices.

However, the use of recycled plastics and metals can also attract environmental and financial incentives from local authorities (e.g., green certificates), lowering the overall operating cost (OpEx). For instance, in an environmental life cycle assessment (LCA), Eriksson and Finnveden [33] reported that efficient use of waste plastics (e.g., as fuel) could also yield a net negative contribution to greenhouse gases (mainly $CO_2$) as compared to landfill disposal.

**Table 2.** Economic analysis of cryogenic cooling of flexible flowlines and umbilical cables.

| | Note | Measured $m_{LN}$ | Cost of $m_{LN}$ Based on 0.16 \$/Kg [1] | Cost of Cooling 1 km Length | Potentials of Materials in 1 km Length [2] |
|---|---|---|---|---|---|
| | | (a) | (b) = (a × 0.16) | (c) = (b × 1000/L) | |
| Flexible flowline | Length = 0.15 m Diameter = 0.14 m | 2 kg | \$0.32 | \$2100 | HDPE: 7 tonnes Carbon steel: 15 tonnes Stainless steel: 10 tonnes Total value > \$35,000 |
| Umbilical cable | Length = 0.15 m Diameter = 0.18 m | 2.5 kg | \$0.4 | \$2700 | HDPE: 6 tonnes Galvanized carbon steel: 22 tonnes Copper: 1 tonne Total value > \$40,000 |

[1] Onsite production price of LN [34]. [2] Prices based on local market [35].

## 4. Conclusions

This paper investigates the feasibility of cryogenic comminution of decommissioned (waste) underwater cables and flowlines. Thermal analyses confirm that the materials in the subsea cables and flowlines can be effectively cooled down to their respective ductile-to-brittle (or glass, for polymers) transition temperatures. Accordingly, as shown by laboratory testing results, the materials were successfully fractured and separated into constituent materials. The basic cost analysis of the cryogenic cooling operation shows that economic potential can be derived from the recovered materials. Although the capital costs were

not included in the economic analysis, the relatively low cost of cooling (as compared to the potential cost of the recovered materials) demonstrates that cryogenic comminution of end-of-life umbilical cables and flowlines could provide cost-effective circular recycling of these materials. Further pilot studies and LCA should be conducted to gauge the complete environmental and economic benefits of the technology.

**Supplementary Materials:** The following supporting information can be downloaded at: https://www.mdpi.com/article/10.3390/su152115651/s1, Figure S1: Variations in average temperature of the flexible flowline (i.e., average of all layers of the flowline during cryogenic cooling); Figure S2: Schematic of umbilical cable cross-section with an outer diameter of 135 mm. Outer sheath: polyethylene; internal plastics: polyamide; filling material: epoxy thermoset; electrical insulation: PVC, etc.

**Author Contributions:** Conceptualization, I.O.; resources and funding acquisition, I.O.; Methodology, I.O. and A.M.; software, A.M.; investigation and formal analysis, A.M. and I.O.; data curation, A.M. and I.O.; writing—original draft preparation, writing—review and editing, I.O. and A.M.; project administration, I.O. All authors have read and agreed to the published version of the manuscript.

**Funding:** This research was funded by Woodside Energy Technology Pty Ltd. (Future-Lab Australia).

**Data Availability Statement:** The data presented in this study are available on request from the corresponding author.

**Acknowledgments:** Ibukun Oluwoye acknowledges financial support from the Japan Society for the Promotion of Science (JSPS), under the FY2022 JSPS Postdoctoral Fellowship for Research in Japan (Standard, P22735).

**Conflicts of Interest:** The authors declare no conflict of interest.

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
