# Peer review of "Cryogenic Comminution of Subsea Cables and Flowlines: A Pathway for Circular Recycling of End-of-Life Offshore Infrastructure"

_sustainability, doi:10.3390/su152115651_

Round 1

Reviewer 1 Report

Comments and Suggestions for Authors

Size and resolution of Figure 3a need to be improved.

Feasibility analysis: The authors need to consider that the treatment does not consume LN only. There are many factors, such as machinery, electricity, technicians, etc. I am not asking to perform the analysis again, but at least this needs to be clear and adequately justified in the text.

Table 2: please remove the extra , in $40,000.

Please re-write the conclusions. Consider providing recommendations regarding industrialization of the process.

Reviewer 2 Report

Comments and Suggestions for Authors

We received paper from sustainability-2637645 with the title “Cryogenic comminution of subsea cables and flowlines: a pathway for circular recycling of end-of-life offshore infrastructure”. The paper have interesting topic and addressed the important aspect related recycle process. Before it goes to further steps, it need to be revised in several points:

11.       In the abstract, please add the most important results in value mode.

22.       In the Fig. 1, it is a original drawing or gathered from previous work?

33.       Related to the recycle process, please update the manuscript by adding previous paper i.e. http://dx.doi.org/10.25103/jestr.151.10

44.       In fig. 3, the detail diameters of each circle were too small. Revised it.

55.       In the materials and methods, please stated simulation was used what FEA software and what edition?

66.       What the author wants to show to the reader related to Fig. 4? There is no urgency to show this fig.

77.       In the study, how the authors crushed the sample was not clear. What machine that used? Add the schematics study with the equipment that used to crush the specimen as shown in Fig. 5

88.       This claim “Our result shows that the component materials can be well sorted and separated.”, how the authors used separator system and sorting system?? Since in the manuscript there is no information related to this issue.

99.       Related to the basic cost, is that the authors only used the materials cost or also incorporate with manufacturing cost of the specimen?

110.   In the conclusion, please stated all the results that occurred during the study and write in the last sentences of the conclusion the used or the future application that can be applied from the present study.

Comments on the Quality of English Language

English are fine

Round 2

Reviewer 2 Report

Comments and Suggestions for Authors

After carefully check the revised manuscript, the present paper can be accepted.

Comments on the Quality of English Language

please check the entire manuscript to fix the grammer error or miss spelling.